# Microbial Shifts Following Five Years of Cover Cropping and Tillage Practices in Fertile Agroecosystems

**DOI:** 10.3390/microorganisms8111773

**Published:** 2020-11-11

**Authors:** Nakian Kim, María C. Zabaloy, Chance W. Riggins, Sandra Rodríguez-Zas, María B. Villamil

**Affiliations:** 1Department of Crop Sciences, University of Illinois, Turner Hall, 1102 S. Goodwin Ave., Urbana, IL 61801, USA; nakhyun2@illinois.edu (N.K.); cwriggin@illinois.edu (C.W.R.); 2Centro de Recursos Naturales Renovables de la Zona Semiárida (CERZOS), Universidad Nacional del Sur (UNS)-CONICET, Ave. de los Constituyentes s/n, Bahía Blanca 8000, Argentina; mzabaloy@uns.edu.ar; 3Department of Animal Sciences, University of Illinois, 30 ASL, 127 W. Gregory Dr., Urbana, IL 61801, USA; rodrgzzs@illinois.edu

**Keywords:** archaea, bacteria, fungi, soil health, biological indicators, agronomy, sustainable agriculture

## Abstract

Metagenomics in agricultural research allows for searching for bioindicators of soil health to characterize changes caused by management practices. Cover cropping (CC) improves soil health by mitigating nutrient losses, yet the benefits depend on the tillage system used. Field studies searching for indicator taxa within these systems are scarce and narrow in their scope. Our goal was to identify bioindicators of soil health from microbes that were responsive to CC (three levels) and tillage (chisel tillage, no-till) treatments after five years under field conditions. We used rRNA gene-based analysis via Illumina HiSeq2500 technology with QIIME 2.0 processing to characterize the microbial communities. Our results indicated that CC and tillage differentially changed the relative abundances (RAs) of the copiotrophic and oligotrophic guilds. Corn–soybean rotations with legume–grass CC increased the RA of copiotrophic decomposers more than rotations with grass CC, whereas rotations with only bare fallows favored stress-tolerant oligotrophs, including nitrifiers and denitrifiers. Unlike bacteria, fewer indicator fungi and archaea were detected; fungi were poorly identified, and their responses were inconsistent, while the archaea RA increased under bare fallow treatments. This is primary information that allows for understanding the potential for managing the soil community compositions using cover crops to reduce nutrient losses to the environment.

## 1. Introduction

Ecological intensification is defined by Bender et al. [1] as an approach “to enhance agricultural sustainability through the promotion of biodiversity and targeted management of soil communities.” Within this framework, cover cropping is one of the main tools that is widely researched and promoted as a promising strategy to mitigate soil nutrient loss, greenhouse gas emissions, and water pollution [2,3,4]. Cover cropping (CC) benefits agroecosystem by improving soil organic matter (SOM) and water retention [5,6], suppressing weeds [7,8], reducing soil erosion [7], and preventing soil nutrient loss by scavenging excess nutrients [9,10,11,12]. The realization of benefits from CC depends heavily on weather conditions and management practices, such as tillage [13,14]. Tillage usually increases the aeration and temperature, as well as the susceptibility to erosion of the topsoil [15]. Likewise, tillage reduces the provision of benefits from CC to soil organic C (SOC) content [16] and soil N [17] by increasing the N-leaching potential of legume CC and reducing the N-scavenging capability of grass CC [18].

Due to their direct and indirect impacts on soil properties and nutrient cycling, CC in combination with tillage practices have the potential to alter the structure and function of the soil microbiome. Microbes are major drivers of the agriculturally and environmentally important soil biogeochemical processes [19,20,21], including those that dictate soil nutrient losses [2,22]. Therefore, understanding how cover cropping and tillage contribute to the soil microbiome is a necessary step toward optimizing the sustainability and efficiency of these practices. At the community level, research syntheses have revealed that CC [7,23] and conservation tillage practices, such as no-till [24], increase the abundance, activity, and diversity of the soil microbiome as a whole, which may promote functional redundancy and resilience against stress.

Community-level inferences, however, only superficially explain how the soil microbiome responds to CC and tillage. New metagenomics research allows for a closer look into the soil microbial composition and function and the changes brought about by agricultural practices [25,26,27]. This usually involves using β-diversity measures to detect shifts in the microbial community structure between samples [28], interpreting the changes with further analyses targeting specific functional genes, such as *amo*A, involved in nitrification [29], and characterizing the compositional shifts that identify the responsive taxa [30,31]. So far, these responsive taxa have been identified at different scales, from phyla [32] to operational taxonomic units (OTUs) [30] or species [33] level. These sensitive taxa may serve as useful indicators for probing the complex microbial behaviors, or even as bioindicators of soil health [26]. Studies on responsive taxa have thus far shown that CC and tillage primarily impose selection pressures on the soil microbes by changing the soil nutrient availability and quality [30,34,35]. Due to the novelty of metagenomics studies in agricultural research, field studies searching for indicator taxa within systems deploying cover cropping and tillage are still scarce. Prior studies either only identified responsive taxa in the samples from CC treatments that showed significant differences in ordination methods [30] or searched for microbes with specific genes, such as those related to residue decomposition [36]. No CC metagenomics study has sought indicator taxa at lower taxonomic ranks without any such constraining conditions.

Therefore, our main goal was to identify potential bioindicators of soil health from microbes that were responsive to CC and tillage treatments under field conditions after five years of treatments. Soil properties, CC biomass, and their C and N contents, along with published reports on the indicator taxa provided the context for our findings and their implications on soil nutrient losses. The results from this field study provide primary information on the soil microbiome within corn–soybean rotations including CC. This information represents a first step toward designing a targeted management strategy of the soil community composition with cover crops to reduce nutrient losses within the fertile conditions that are typical of prime agricultural landscapes.

## 2. Materials and Methods 

### 2.1. Experimental Site Description and Management Practices

The experimental site was established in the fall of 2012 at the Crop Sciences Research and Education Center at Urbana, IL, USA (40°03′25.2″ N 88°13′37.2″ W), as part of a larger CC and tillage study [13,14]. The site was level to gently sloping (from 0 to 2%) and spanned a Drummer (70%)–Catlin (10%)–Flanagan (20%) soil association that was dominated by fine-silty, superactive Mollisols [37].

Treatments of tillage and corn–soybean CC rotations were arranged in a split-block design with eight blocks in total. Four blocks were each assigned to corn and soybean, which were rotated annually. Tillage was randomized into each block in the north–south direction into tilled (T) and no-till (NT) plots, and subplots of CC were randomly allocated in the west–east direction. The CC treatments included annual ryegrass (*Lolium multiflorum* Lam.) before and after both cash crops (CarSar), cereal rye (*Secale cereale* L.) after corn and hairy vetch (*Vicia villosa* Roth.) after soybean (CcrShv), and bare fallows before and after corn and soybean as control treatments (CT). In any given year, each CT and CarSar treatment had eight subplots, four under the cereal rye phase of CcrShv and four under the hairy vetch phase of CcrShv. Our results are presented and discussed as the average for a given corn–soybean CC rotation to reflect the overall impact of each management strategy.

Detailed information regarding the soil properties, crop yields, and field management practices during the project period from 2012 to 2017, is publicly available [38]. Briefly, corn and soybean cash crops were planted around May and harvested between October to November each year. Pre-plant N fertilizer was applied to corn as urea ammonium nitrate (UAN 28%) at the rate of 190 kg N ha^−1^. Plots under the tillage treatment were chisel-plowed down to 20–25 cm deep in the spring between CC suppression and cash crop planting. The CC seeds were broadcasted by hand on standing cash crops in September. Seeding rates were 16.8 kg ha^−1^ for annual ryegrass, 22.4 kg ha^−1^ for hairy vetch, and 100 kg ha^−1^ for cereal rye. Cover crops were suppressed with glyphosate (N-(phosphonomethyl)glycine) at 1.12 kg a.i. ha^−1^ by the end of April. The exact dates of the field management practices are shown in Appendix A.

### 2.2. Soil and Biomass Sampling and Procedures

Soil samples were collected on 21 April 2017 following five years of treatments in place. An Eijelkamp grass plot sampler (Eijkelkamp Agrisearch Equipment, Giesbeek, The Netherlands) was used to take two composited subsamples of 500 g each per subplot to a depth of 10 cm for soil DNA analyses. Soil samples were kept on ice in the field and frozen once in the laboratory. Three soil core samples with a diameter of 4.3 cm were also taken randomly down to a 30 cm depth for each subplot using a tractor-mounted automated soil sampler (Amity Technology, Inc., Fargo, ND, USA). Soil nitrate-N (NO_3_-N) and ammonium-N (NH_4_-N) (mg kg^−1^) were measured using these moist field soil samples using KCl extraction (1:5 ratio) and analyzed using a SmartChem 200 Discrete Analyzer Auto-Spectrophotometer (Westco Scientific Instruments, Inc., Brookfield, CT, USA). Soil samples were then air-dried and sieved to 2 mm and sent to a commercial laboratory (Brookside Laboratories, Inc., New Bremen, OH, USA) that used standard methods that are recommended for the U.S. Central Region [39] for the determination of the cation exchange capacity (CEC, cmol kg^−1^), soil pH, available soil P (Bray P I extraction), and soil organic matter (SOM, g kg^−1^) using loss on ignition. Cover crop biomass samples were taken on 11 April 2017 at the time of spring growth suppression using three random tosses of 0.25 m^2^ quadrat per subplot, where they were cut at ground level. The dry weight (DW, g) was recorded after oven drying at 60 °C and the samples were sent to the above commercial lab for the determination of tissue C and N contents (g kg^–1^) and their C:N ratios.

Soil DNA was extracted from 0.25 g of the composited soil samples using PowerSoil^®^ DNA isolation kits (MoBio Inc., Carlsbad, CA, USA), following the manufacturer’s instructions. The quantity and quality of the extracted DNA were tested using a Nanodrop 1000 Spectrophotometer (Thermo Fisher Scientific, Waltham, MA, USA) and stored at −20 °C. An Illumina-HiSeq-compatible amplicon library containing individual barcodes for each sample was constructed as described in Sun et al. [26]. The library prep was accomplished using 25 µL volumes with a 1× buffer (GoTaqfi Flexi buffer; Promega Corp., Madison, WI, USA), 2.5 mM MgCl_2_, 200 µM dNTPs, 0.4 µM of each primer (forward and reverse), 1.0 µL template DNA (pooled amplicons), and 1.0 unit of GoTaq polymerase. PCR was performed using a BioRad T100 thermal cycler with the following parameters: initial denaturation at 95 °C for 10 min, followed by 34 cycles of amplification (45 s at 95 °C, 45 s at 58 °C, 45 s at 72 °C), and a final extension at 72 °C for 10 min. PCR products were visualized on a 1.3% agarose gel containing GreenGlo™ Safe DNA dye (Denville Scientific, Inc., Metuchen, NJ, USA) under UV illumination. Bacterial 16S rRNA gene (V4 region) was amplified using a primer set of 515F (GTGYCAGCMGCCGCGGTAA) and 806R (GGACTACVSGGGTWTCTAAT) [40], archaeal 16S using 349F (GTGCASCAGKCGMGAAW) and 806R (GGACTACVSGGGTATCTAAT) [41], and a fungal ITS (internal transcribed spacer) region using 3F (GCATCGATGAAGAACGCAGC) and 4R (TCCTCCGCTTATTGATATGC) [42]. The primers were designed as a 5′-PCR-specific + gene region + 3′-PCR-specific + 10 nt barcode and the Fluidigm platform used two primer sets simultaneously to create the final DNA amplicon. A Qubit Fluorometer (Thermo Fisher Scientific, Waltham, MA, USA) quantified the resulting amplicon libraries, which were then run on a Bioanalyzer (Agilent, Santa Clara, CA, USA) to evaluate the profile of fragment lengths. The barcoded libraries were pooled in equimolar concentrations and diluted to 10 nM. The diluted libraries were sequenced at the Roy Carver Biotechnology Center Functional Genomics lab at the University of Illinois at Urbana-Champaign (Urbana, IL, USA) using paired-end sequencing on the Illumina MiSeq nano 2 (Illumina, San Diego, CA, USA), yielding 250 nt long reads.

### 2.3. Bioinformatics Analysis

Quality checking and processing of the sequences were done through QIIME2 [43,44]. The demultiplexed sequences were filtered using a Q score of 30 [45], resulting in retaining bacterial sequences between base-pair positions 6 to 250, fungal sequences 6 to 200, and archaeal sequences 6 to 136. Next, sequences were denoised by removing chimeric and low-quality sequences with the chimera method consensus option in plugin DADA2 [46]. Sequences were aligned with Mafft [47] to then create the phylogenetic tree using FastTree [48]. Reference sequences from the SILVA ribosomal RNA gene database (silva-132-99-515-806-nb-classifier_2019_4) [49] were used to compare bacterial and archaeal 16S rRNA sequences, and Fungi_97_classifier_2019_4 was used for ITS sequences, which were then clustered them into OTUs at 97% similarity threshold. The rarefaction curves plateaued at the sampling depths of 5000 sequences per sample for bacteria, 900 for fungi, and 300 for archaea (Appendix A). At these depths, QIIME2 calculated the number of observed OTUs, Shannon’s Diversity Index (H’), and the Chao 1 Richness Index (Chao1) for each sample. The weighted UniFrac distance was calculated using QIIME2 to measure the β-diversity. The rarefaction curves (Appendix A) were created using package ggplot2 in R version 3.5.3 [50,51].

### 2.4. Statistical Analysis

The relative abundances (RAs, %) of each OTU were statistically analyzed to identify the responsive microbes and estimate the treatment effects. The initial set of responsive microbes were selected using a bootstrap forest partitioning method with the JMP^®^ predictor screening platform [52,53,54]. This led to a selection of 42 out of 1832 OTUs for bacteria, 5 out of 19 for archaea, and 36 out of 313 for fungi that contributed at least one percent to the variability in the model algorithms (Appendix A). With only five OTUs selected, further selection was unnecessary for archaea. Principal component analyses (PCAs), were used on the pool of top contributing bacterial and fungal OTUs to further remove redundancy and avoid issues of multicollinearity. Using the FACTOR procedure in SAS software version 9.4 (SAS Institute, Cary, NC) with priors = 1, the RAs of these OTUs were summarized into a set of uncorrelated composite variables, or principal components (PCs). PCs with eigenvalues ≥ 1 that also explained at least 5% of the variability in the data set were used as independent variables for further analysis. OTUs with a significant correlation with each PC (PC loading value > |0.5|) were considered responsive microbes [55]. Linear mixed model analyses were fit using the GLIMMIX procedure in SAS software to each of the response variables: soil and CC biomass properties, α-diversity measures, and PC scores of top contributing OTUs. Tillage and CC were considered fixed effects, whereas blocks were considered random terms in the analyses of variance. When appropriate, least-square means of the response variables were separated by treatment levels, using the pdiff option and setting the probability of a type I error at α = 0.10.

The ggplot2 package in R version 3.5.3 was used to plot the RA responses for each significant relationship between the PC scores and treatments [50,51]. Figures 1–3 and Appendix A are visual representations of the combined results of the PCA and their mean separation procedure showing the most responsive bacterial and fungal indicator OTUs based on their effects on the RAs following five years of treatments within corn–soybean rotations (a complete list of indicators within each PC for each taxon is shown in Appendix A). For each taxon, the response of each OTU to tillage (Figure 1), to CC (Figure 2), or their interaction was calculated as the mean PC score for a given treatment multiplied by the PC loading score of the listed OTU. Likewise, the whiskers represent the standard error of the mean PC scores for each treatment multiplied by the absolute value of each OTU loading. The β-diversity, measured using the weighted UniFrac distance, was analyzed with pairwise PERMANOVA (permutational multivariate analysis of variance) by QIIME2 to compare the differences between treatment levels using pseudo-F test statistics and their *p*- and *q*-values, which are the expected false positive and negative rates, respectively, in multiple hypothesis testing [56,57].

## 3. Results

### 3.1. Soil and Cover Crop Biomass Properties

Table 1 shows the mean treatment values and their standard errors, along with the results of the mean separation procedures for the soil and CC biomass properties in response to the imposed treatments. The soil properties of CEC, soil pH, SOM, NH_4_-N, and available P did not differ statistically among CC or between the tillage treatment levels (*p* > 0.1). Only the soil NO_3_-N showed a marginal statistically significant CC effect (*p* = 0.07), where it was less under CarSar compared to CcrShv and CT; NO_3_-N did not display significant tillage effects (*p* > 0.1). The biomass C was statistically greater under no-till than chisel tillage (*p* = 0.04) and marginally statistically greater for Ccrshv than CarSar (*p* = 0.08). The biomass N content, C:N ratio, and dry weight did not differ among CC treatments or between the tillage practices (*p* > 0.1).

### 3.2. Overall Characterization of the Soil Microbiome

The bacterial community had more than 1.2 million 16S V4 region sequences clustered into 1832 different OTUs. The archaeal community had 13,272 archaeal 16S rRNA region sequences clustered into 19 OTUs. The fungal community had 213,860 ITS region sequences clustered into 313 OTUs. The α-diversity of bacteria, archaea, and fungi measured as OTUs, H’, and Chao 1 all did not statistically differ between CC and between tillage treatment levels (*p* > 0.1) (Table 2). 

As for the β-diversity measure (Table 3), the bacterial community structure statistically differed between tillage practices but not between CC treatments. The archaeal community structure did not differ statistically between tillage treatments; the community structure of CarSar and CcrShv did differ statistically from that of CT, but not between CarSar and CcrShv. The fungal communities showed significant shifts in the community structure between tillage practices and between CC treatments.

The most abundant bacterial phyla across the samples were Proteobacteria (34.2%), followed by Actinobacteria (20.4%), Chloroflexi (9.7%), Acidobacteria (9.5%), and Bacteriodetes (8.5%). The archaeal community was dominated by phylum Thaumarchaeota (96.8%) and class Nitrososphaeria (95.8%). The most abundant fungal phylum was Ascomycota (54.8%), followed by Basidiomycota (9.5%), and Mortierellomycota (4.6%). Appendix A shows the mean treatment values for the PC scores of the RAs of microbial phyla in response to the tillage treatments. Only the bacterial phyla Acidobacteria, Bacteroidetes, Chloroflexi, Gemmatimonadetes, and Proteobacteria showed a significant statistical tillage effect (*p* = 0.002), where Bacteroidetes and Proteobacteria were more abundant under chisel tillage than no-till, while the other three responded in an opposite manner. The RAs of the other bacterial, archaeal, and fungal phyla did not differ statistically between CC treatments or between tillage practices (Appendix A).

### 3.3. OTUs Responses to Cover Crop Rotations and Tillage Treatments

#### 3.3.1. Bacteria

A total of seven PCs explained 53.4% of the variability in the 42 selected top-contributing bacterial OTUs, yet only PC1 to PC5 had significant correlations (loadings ≥ |0.5|) with their conforming OTUs (Table 4 and Appendix A).

PC1 explained 14% of the variability, including the positive loadings from five OTUs belonging to genera *Cellulomonas*, *Solirubrobacter*, *Altererythrobacter*, *Massilia*, and the family AKYH767, and four negative loadings each for OTUs in the order SBR1031, families A21b and Gemmatimonadaceae, and the genus *Elin6067*. PC2 explained an additional 9.5% of the variability of the data set and had positive loadings from two OTUs each belonging to the genus *Reynarella* and the class Gitt-GS-136, and negative loadings for two OTUs belonging to the order C0119 and the family JG30-KF-A59. PC3 showed positive loadings from OTUs in the family Opitutaceae and the species *Nitrospira japonica*. PC4 had a significant negative loading from an OTU in the family A4b, whereas PC5 showed a positive loading from an OTU belonging to the phylum Latescibacteria. While PC6 and PC7 each explained an additional 5% of the variability found in the input data set, neither had OTUs with PC loadings > |0.5|.

ANOVA on these bacterial PCs detected marginally significant tillage (*p* < 0.1) and significant CC (*p* < 0.01) main effects from PC1, PC2, and PC3 (Table 4; Figure 1 and Figure 2).

#### 3.3.2. Fungi and Archaea

Six PCs explained 44.7% of the variability represented by the 36 selected top-contributing fungal OTUs (Appendix A). PC1 had an eigenvalue of 3.2 and explained 9.1% of the variability, including a significant positive loading from *Saitozyma podzolica*. PC2 explained an additional 8.6% of the variability with a negative loading of an OTU belonging to class Agaricomycetes. PC3 included a significant positive loading from the genus *Alternaria* and a negative loading from the genus *Penicillium*. PC4 included a positive loading from an OTU in the class Agaricomycetes, which was different from that of PC2. PC5 had a positive loading from an OTU in the order Agaricales, whereas PC6 had a significant positive loading from genus *Trichoderma*.

The ANOVA on these fungal PCs showed that the tillage main effects were significant (*p* < 0.01) for PC2 and marginally significant (*p* < 0.1) for PC3 and PC5 (Table 4; Figure 1 and Figure 2). The main CC effects were significant for PC3 (*p* < 0.05) and PC4 (*p* < 0.01). The CC and tillage interaction effect was significant (*p* < 0.05) for PC5 (Figure 3). The mean separation procedure on these PCs showed that the mean PC scores were statistically greater with chisel tillage than no-till for PC2 and PC3, but the opposite was true for PC5. For PC3, the mean PC score was statistically greater with CarSar than CT and CcrShv. For PC4, the mean PC scores were statistically different for all CC treatments in the order of CarSar, CcrShv, and CT from greatest to smallest. For the CC and tillage interaction effect, the mean PC score was statistically greater with NT × CcrShv than NT × CarSar, T × CT, and T × CcrShv, while the other interactions took intermediate values.

The five top-contributing archaeal OTUs all belonged to the family Nitrososphaeraceae: an unidentified *Candidatus nitrososphaera* OTU and unidentified archaeal OTUs SCA1154, SCA1158, SCA1166, and SCA1173 (Appendix A). The ANOVA on each of these OTUs detected only one statistically significant main effect of CC (*p* < 0.05) on SCA1173 (Table 5). The mean RA of SCA1173 was statistically higher under bare fallow (CT) than under CarSar, with the rotation CcrShv showing intermediate values.

## 4. Discussion

Unlike the shifts in the soil microbiome we detected, chemical soil properties were only marginally influenced by cover crop rotations and tillage practices, partially because of the innate buffering capacity of the rich Mollisols across the region [13,14,58]. Soil NO_3_^−^ was the only soil chemical property that responded to the treatments, and our results are consistent with past reports stating that grass CC reduces soil NO_3_^−^ more effectively than legume CC that fixes N for themselves [12,59]. This effect, however, was less pronounced for NH_4_^+^, perhaps because crops preferentially utilize NO_3_^−^ over NH_4_^+^ [60]. Similar to our results, there is no strong evidence that changes to the pH by CC and tillage can be meaningful enough for the soil microbiome [61,62]. Moreover, the high CEC of the present study’s soil indicated a greater buffering capacity against pH changes (Table 1). This innately high CEC would be resilient against changes from CC and tillage [13]. Indeed, a past meta-analysis on CC effects on soil properties reported smaller percentage changes in CEC than those of our results, but the differences in CEC were still not statistically significant [63]. Likewise, contributions to SOM from CC were likely overshadowed by those from cash crops [10]. Regarding the available P levels, the lack of response to the treatments that was found in our study somewhat agrees with the findings of the meta-analysis of Daryanto et al. [7], which concluded that successful soil P loss reduction by CC needs to be accompanied by other practices, such as reducing P fertilization and diversifying CC species. Similarly, due to their early spring suppression time, which ensures high tissue quality, rotations including CC did not show any important contrast in tissue composition, as measured by their C:N ratios.

On the other hand, the indicator microbes found in this study showed responses that were consistent with their known characteristics. This consistency, however, was not reflected by the β-diversity and phylum-level analyses, which highlighted the importance of finer-scale taxonomic analysis in metagenomics studies. Most notably, the RAs of bacterial OTUs consistently differed between the bare fallow CT and the legume-grass CC rotation CcrShv (Figure 2). Therefore, these bacterial OTUs could be categorized into two groups, which were “CT-favored” and “CcrShv-favored,” and each had distinct characteristics that could explain this dichotomy. The seven CcrShv-favored bacteria were likely aerobic organotrophs: the genera *Cellulomonas* [33,64], *Solirubrobacter* [65], *Altererythrobacter* [66], *Massilia* [67], the family AKYH767 [68,69], and two in the class Ktedonobacteria [70,71]. As CCs provide substrates as crop residues and root exudates, these decomposer OTUs responding positively to CC treatments was expected [22,72]. Furthermore, the extra supply of N fixed by the hairy vetch phase of the CcrShv would have furthered the CC benefits on these indicator OTUs than CarSar [73]. Moreover, all CcrShv-favored OTUs, except for the two Ktedonobacteria, belonged to the phyla that Fierer et al. [74] proposed, as the copiotrophs adapted to the nutrient-rich environment. Therefore, these CcrShv-favored OTUs represent the copiotrophic guild and could be used as bioindicators to gauge the microbiome’s capacity to decompose labile organic matter.

On the other hand, characteristics of the eight CT-favored bacteria implied an adaptation to stressful or oligotrophic soil environments. Each of the three CT-favored OTUs in the order SBR1031, and the families Opitutaceae and A21b were either strict or facultative anaerobes [75,76,77]. The OTU in the family Gemmatimonadaceae was documented to have adapted to lower soil moisture [78]. The OTU in the class Gitt-GS-136 was not well documented. Most importantly, the CT-favored bacteria included an ammonia-oxidizing bacterium (AOB; Nitrosomonadaceae) [79], a nitrite-oxidizing bacterium (NOB; *N. japonica*) [80], and a potential nitrate-reducing bacterium (Reyranella) [81]. As autotrophs that gain energy from oxidizing N compounds, AOB and NOB are better adapted to oligotrophic conditions compared to organotrophs by using organic C [82]. Similar to our results, Zheng et al. [36] found that the AOB family Nitrosomonadaceae decreased in abundance under legume CC, while [83] found that the abundance of *N. japonica* had a negative correlation with soil SOC. The reported characteristics of these CT-favored OTUs are consistent with the bare fallow soils being more oligotrophic [84], anaerobic, and drier [6,85] than the soil under CC. Therefore, these CT-favored bacteria can be used as bioindicators of these soil properties, the oligotrophic microbial guild, and the nitrifying and denitrifying potentials of the soil microbiome.

Bacterial responses to tillage treatments were only marginal (*p* < 0.1) but still displayed consistency with their known characteristics (Figure 1). In PC1, the CcrShv-favored OTUs were also more abundant with chisel tillage, while the CT-favored OTUs were more abundant with no-till conditions (Table 4). Tillage breaks the crop residues and incorporates them into the soil while increasing the surface temperature and aeration, thereby facilitating decomposition [86,87]. Under these conditions, the CcrShv-favored copiotrophic and aerobic organotrophs are expected to be more competitive over CT-favored stress-tolerant oligotrophs [34,74]. However, this pattern did not hold in PC2 and PC3, as CcrShv-favored OTUs were more abundant under no-till instead of chisel tillage. Considering that OTUs in PC2 and PC3 were either aerobic or facultatively anaerobic [70,71,75,80,88], the CT-favored OTUs in these PCs might have been responding positively to the soil aeration from tillage. The two Ktedonobacteria OTUs that responded positively to no-till somewhat contradicted a past report, where this class increased with conventional tillage over reduced tillage [89]. However, the class Ktedonobacteria is not well studied, and our results, which showed that its OTUs behaved uniquely from other copiotrophs (Figure 1 and Figure 2), suggest that more studies should focus on this class. The tillage effects on the soil microbiome found in this study were somewhat ambiguous compared to those of CC, as demonstrated by the marginal statistical difference (Table 4). This may owe to chisel tillage being less aggressive than typical conventional tillage methods, thus the microbial responses to tillage could be more clear-cut under practices like moldboard plowing [24]. The CT-favored oligotrophs in PC2 and PC3 responding positively to chisel tillage contradicts past studies [27,90], yet these analyses were limited to the phyla level, thereby highlighting the discrepancy between the collective microbial behaviors at higher taxonomic ranks and those of individual microbes.

Fungal indicator OTUs or their combined behaviors did not show consistent trends in their responses to CC or tillage (Figure 1 and Figure 2). This variability in fungal responses might be a result of a diverse fungal ecology that includes saprotrophs, symbiotrophs, and pathogens [91]. Indeed, the two fungal OTUs in PC3—Alternaria and Penicillium—that responded to both CC and tillage methods are common crop endophytes, which contrasted with our bacteria OTUs that are unlikely plant symbionts [92]. Possibly, fungal OTUs that responded positively to no-till could be hyphal species that are negatively affected by the physical disruption from tillage [35]. Some fungi are more specialized at degrading recalcitrant parts of the crop residues and SOM than bacteria [93]. Fungi may therefore be more competitive than bacteria with CarSar, as grass-only CarSar provides less inorganic N for bacteria to degrade recalcitrant substrates. This could explain why fungal responses to CarSar differed statistically from CT and CcrShv (Figure 2).

The one archaeal indicator SCA1173 that responded significantly to CC belonged to the family Nitrososphaera, one of the most represented ammonia-oxidizing archaea (AOA) [94,95]. Interestingly, this indicator archaeon was more abundant with CT than CarSar, with an intermediate response to CcrShv (Table 5). Zhalnina et al. [96] found that an uncultured Nitrososphaera OTU positively correlated with the soil ammonia (NH_3_) level, which is the substrate for ammonia oxidation. Considering that CT does not have CC to compete for soil inorganic N and CcrShv fixes NH_3_, AOA responding negatively to CarSar was expected. Moreover, the oligotrophic conditions of CT provide a better niche for autotrophic AOA, which further explains the statistical difference between its responses to CT and CarSar [82]. Considering the small number of total archaeal OTUs identified in this study, current methods should be improved to detect archaea as potential bioindicators of soil health.

Overall, our results show that cover cropping and tillage benefit copiotrophic decomposers, which was more pronounced with legume-grass CC rotation than grass-only CC. Because legume CC can fix a considerable amount of N until suppression, copiotrophic decomposers compete less with the microbes for soil inorganic N than grass CC [12,13,97]. Under this N surplus, copiotrophs dominate the decomposition and promote a positive priming effect that decomposes SOM and crop residues [98,99]. Here, the low C:N of the CC residues at the time of suppression would lead to N mineralization [100]. This effect manifested as similar levels of total soil inorganic N between the CC treatments (Table 1) [101]. The higher microbial N mineralization could be partially responsible for the statistically higher NO_3_- level with CcrShv than CarSar, explaining the reduced ability of legume CC to scavenge N compared to grass CC [102,103].

The bacterial community structure differed significantly by tillage only (Table 3), unlike our OTU-level results that showed sensitivity to CC treatments (Figure 1 and Figure 2). Instead, the β-diversity results were comparable to the bacterial phyla responding significantly to tillage (Appendix A), thereby failing to represent the significant behaviors of the bioindicator OTUs. Unlike bacteria, the responses of indicator archaeal and fungal OTUs differed statistically between the CC and tillage treatments, which also showed significant shifts in the community structure (Table 4 and Table 5; Figure 1 and Figure 2). Compared with the 1832 total bacterial OTUs, fungi and archaea only had 313 and 19 OTUs, respectively. With a much lower species richness, each fungal and archaeal OTU had much more of a statistical impact on the β-diversity, which is likely the reason why β-diversity followed the trends of archaeal and fungal OTUs. This discrepancy between the analysis at finer taxonomic scales and at the community scale demonstrated that limiting the analysis to the latter may neglect valuable information regarding the soil microbiome, especially considering the striking consistency between the behaviors of indicator bacterial OTUs.

## 5. Conclusions

This study showed that cover cropping and tillage differentially changed the RAs of the copiotrophic and oligotrophic guilds, mainly through changing the soil nutrient availability, aeration, and moisture conditions. The legume-grass CC rotation benefited the decomposers noticeably more than grass-only CC, likely due to N-fixation. We identified a handful of sensitive taxa that are potential bioindicators of important soil processes. Among them, the promising bioindicators were bacterial genera *Cellulomonas*, *Solirubacter*, *Altererythrobacter*, and *Massilia* representing the aerobic decomposers, bacterial family Nitrosomonadaceae for ammonia oxidation, and bacterial genera *Nitrospira* and *Reyranella* for nitrite oxidation and nitrate reduction, respectively. Unlike bacteria, we found fewer indicator fungi and archaea; fungi were poorly identified and their responses were inconsistent, while the AOA family Nitrososphaeraceae responded positively to CT. Future research should verify and expand on the bioindicators found in this study under different temporal and geographical conditions to develop them into a valuable soil health evaluation tool. Our results also showed that community-level or phyla-level characterization of the soil microbiome may not reflect the microbial behaviors well at lower taxonomic ranks and overlook their significance. This study established the first platform in cover crop research for finer taxonomic analysis on the soil microbiome, and we provided microbial evidence that improves our mechanistic understanding of why grass CC has a higher potential than legume CC within corn–soybean rotations to reduce soil nutrient loss.

## Figures and Tables

**Figure 1 microorganisms-08-01773-f001:**
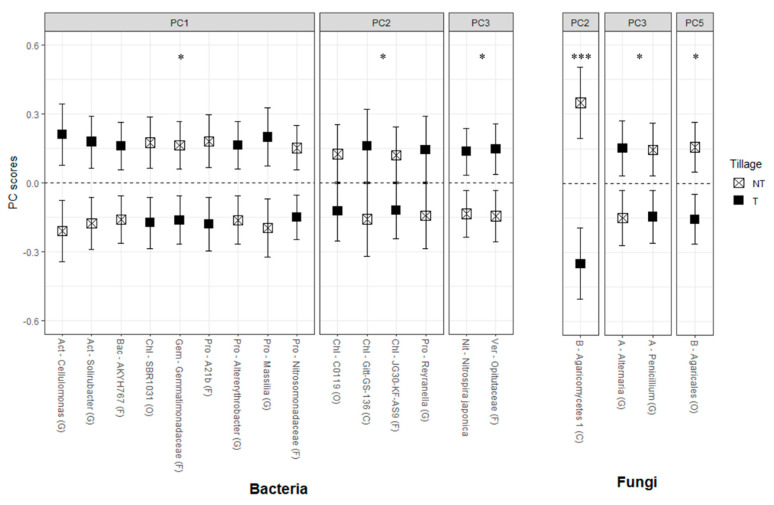
Response pattern of the bacterial and fungal indicator OTUs to tillage. Visual representation of the combined results of the principal component analyses (PCAs) and their mean separation procedure showing the most responsive bacterial and fungal indicator OTUs based on their effects on the relative abundances following five years of tillage within corn–soybean rotations. For each taxon, the response of each OTU to tillage treatments (chisel tillage (filled square), no-till (crossed square)) was calculated as the mean PC score multiplied by the PC loading score of a given OTU. Likewise, the whiskers represent the standard error of the mean PC scores for each tillage treatment multiplied by the absolute value of each OTU loading. The *x*-axes show the three-letter acronym of the OTU’s phylum, followed by the name of the OTU’s lowest identified taxonomic rank and that rank in parentheses. Act, Actinobacteria; Bac, Bacteroidetes; Chl, Chloroflexi; Gem, Gemmatimonadetes; Pro, Proteobacteria; Nit, Nitrospirae; Ver, Verrucomicrobia; A, Ascomycota; B, Basidiomycota; C, Class; O, Order; F, Family; G, Genus. *, ***, significant at the 0.1 and 0.01 probability levels, respectively.

**Figure 2 microorganisms-08-01773-f002:**
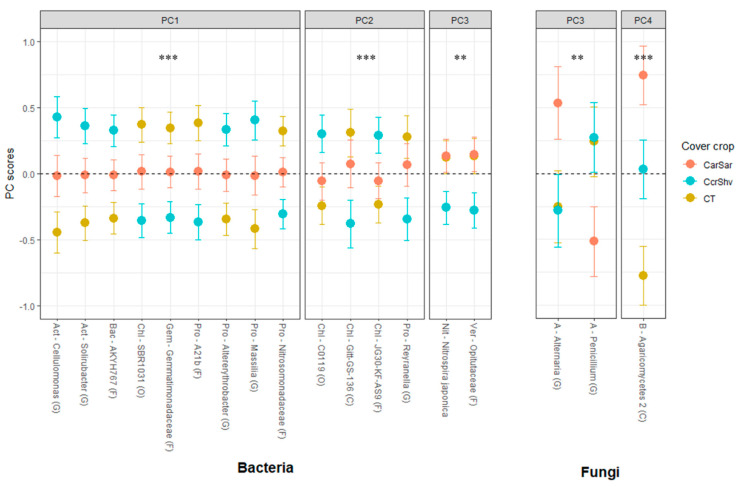
Response pattern of the bacterial and fungal indicator OTUs to cover cropping. Visual representation of the combined results of the principal component analyses (PCAs) and their mean separation procedures showing the most responsive bacterial and fungal indicator OTUs based on Table 4. for PC5 (Figure 3) and PC6. The mean separation procedures showed that the mean PC scores were greater with chisel tillage for PC1, PC2, and PC3. The mean PC score was statistically smaller with CT than CarSar and CcrShv for PC1, but that of CcrShv was smaller than CT and CarSar for PC2 and PC3. For PC5, the mean PC score of NT × CcrShv was statistically greater than T × CcrShv, with all other CC and tillage interactions being intermediate between the two. For PC6, the mean PC scores of T × CT and NT × CarSar were statistically greater than those of NT × CcrShv and T × CarSat, while NT × CT and T × CcrShv took intermediate values. **, ***, significant at the 0.05 and 0.01 probability levels, respectively.

**Figure 3 microorganisms-08-01773-f003:**
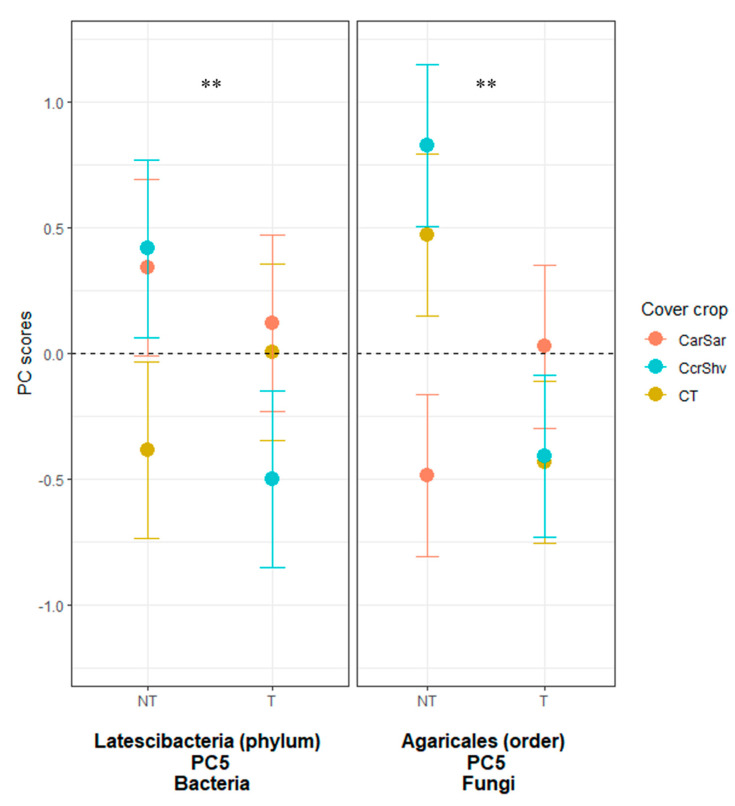
Response pattern of the bacterial and fungal indicator OTUs to cover cropping and tillage. Visual representation of the combined results of the principal component analyses (PCAs) and their mean separation procedure showing the most responsive bacterial and fungal indicator OTUs based on their effects on relative abundances following five years of cover cropping and tillage within corn–soybean rotations. For each taxon, the response of each OTU to cover crop rotation and tillage treatments was calculated as the mean PC score multiplied by the PC loading score of a listed OTU. Likewise, the whiskers represent the standard error of the mean PC scores for each cover crop and tillage treatment multiplied by the absolute value of each OTU loading. Cover crop rotations: CarSar—annual ryegrass CC following both corn and soybean (red), CcrShv—cereal rye following corn and hairy vetch following soybean (blue), CT—bare fallow control (yellow). The responses to cover cropping treatments are grouped by tillage practices on the *x*-axes; NT—no-till, T—chisel tillage. **, significant at the 0.05 probability level.

**Table 1 microorganisms-08-01773-t001:** Treatment mean values and standard errors of the mean (in parentheses) of soil properties and cover crop biomass variables, which were measured after five years of tillage and cover crop rotation treatments. The soil properties included the cation exchange capacity (CEC, cmol kg^−1^), soil pH (pH), and the levels of soil organic matter content (SOM, g kg^−1^), nitrate (NO_3_-N, mg kg^−1^), ammonium (NH_4_-N, mg kg^−1^), and available phosphorus (P, mg kg^−1^). The cover crop biomass variables listed are the contents of biomass carbon (C, g kg^−1^) and nitrogen (N, g kg^−1^), their ratio (C:N), and the biomass dry weight (DW, Mg ha^−1^). For each treatment and within a given row, treatment means followed by similar lowercase letters were not statistically different (α = 0.05).

	Tillage ^†^			Cover Crop ^‡^			
Soil Properties	NT	T	*p*-Value	CT	CarSar	CcrShv	*p*-Value
CEC	21.78 (2.49)	24.04 (2.49)	0.43	24.07 (2.57)	21.59 (2.57)	23.06 (2.57)	0.63
pH	5.95 (0.16)	5.90 (0.16)	0.72	5.96 (0.16)	5.94 (0.16)	5.88 (0.16)	0.74
SOM	37.6 (1.50)	38.3 (1.50)	0.34	37.3 (1.50)	38.1 (1.50)	38.5 (1.50)	0.38
NO_3_-N	1.52 (0.23)	1.47 (0.23)	0.85	1.83a (0.33)	0.85b (0.33)	1.80a (0.33)	0.07
NH_4_-N	12.62 (0.81)	11.90 (0.81)	0.5	11.84 (1.06)	12.68 (1.06)	12.27 (1.06)	0.85
P	5.42 (1.51)	5.79 (1.51)	0.74	6.38 (1.56)	4.31 (1.56)	6.13 (1.56)	0.18
**Cover Crop Biomass**							
C	427.7a (5.00)	414.6b (4.70)	0.04		413.8b (5.30)	428.5a (5.90)	0.08
N	29.3 (1.60)	26.5 (1.50)	0.14		27.9 (1.60)	27.9 (1.70)	1.00
C:N	15.43 (0.71)	15.91 (0.66)	0.58		15.24 (0.69)	16.10 (0.78)	0.41
DW	1.80 (0.10)	1.93 (0.09)	0.27		1.81 (0.09)	1.92 (0.10)	0.33

^†^ NT, no-till; T, chisel tillage. ^‡^ CT, bare fallow control; CarSar, annual ryegrass following both corn and soybean; CcrShv, cereal rye following corn, hairy vetch following soybean.

**Table 2 microorganisms-08-01773-t002:** Mean values and standard errors of the mean (SEM) for the α-diversity parameters of the observed operational taxonomic units (OTUs), Chao 1 Richness Index (Chao1), and Shannon’s Diversity Index (H’) for bacteria, fungi, and archaea taxa following five years of tillage and cover crop rotation treatments.

		OTUs			Chao1			H’		
Taxa	Treatment ^†^	Mean	SEM	*p*-Value	Mean	SEM	*p*-Value	Mean	SEM	*p*-Value
Bacteria	T	206.04	10.14	0.54	206.35	10.19	0.54	7.3	0.07	0.57
	NT	210.11			210.47			7.33		
	CarSar	202.46	10.62	0.38	202.72	10.68	0.37	7.27	0.08	0.27
	CcrShv	213.62			214.2			7.37		
	CT	208.15			208.31			7.32		
Fungi	T	32.94	1.52	0.26	32.97	1.55	0.28	4.19	0.08	0.11
	NT	30.37			30.45			4.01		
	CarSar	31.95	1.56	0.89	31.97	1.58	0.9	4.26	0.12	0.3
	CcrShv	31.9			31.96			4.02		
	CT	31.11			31.2			4.03		
Archaea	T	11.6	0.65	0.58	11.6	0.65	0.58	3.24	0.08	0.73
	NT	11.16			11.16			3.2		
	CarSar	11.45	0.72	0.98	11.45	0.73	0.97	3.24	0.09	0.76
	CcrShv	11.41			11.43			3.24		
	CT	11.29			11.26			3.17		

^†^ Tillage: NT, no-till; T, chisel tillage. Cover crops: CT, bare fallow control following both corn and soybeans; CarSar, annual ryegrass following both corn and soybean; CcrShv, cereal rye following corn, hairy vetch following soybean.

**Table 3 microorganisms-08-01773-t003:** Community structure (β-diversity) measures for bacteria, fungi, and archaea sampled following five years of tillage (Till) and cover crop (CC) treatments, based on pairwise PERMANOVA on weighted UniFrac distances. The pseudo-F measures the significance of the UniFrac distance between the two treatment levels compared, while the *p*-value and *q*-value indicate the probability of type I and type II errors associated with each comparison, respectively.

Taxa	Treatment	Levels Compared ^†^	Sample Size	Pseudo-F	*p*-Value	*q*-Value
Bacteria	Till	NT–T	142	4.33	0.001	0.001
	CC	CT–CarSar	95	1.08	0.296	0.444
		CT–CcrShv	94	1.30	0.192	0.444
		CarSar–CcrShv	95	0.77	0.774	0.774
Fungi	Till	NT–T	129	2.78	0.019	0.019
	CC	CT–CarSar	83	5.42	0.002	0.003
		CT–CcrShv	87	3.14	0.013	0.013
		CarSar–CcrShv	88	4.11	0.001	0.003
Archaea	Till	NT–T	130	1.12	0.272	0.272
	CC	CT–CarSar	86	2.90	0.054	0.081
		CT–CcrShv	87	3.98	0.01	0.03
		CarSar–CcrShv	87	0.80	0.437	0.437

^†^ Tillage: NT, no-till; T, chisel tillage. Cover crops: CT, bare fallow control following both corn and soybeans; CarSar, annual ryegrass following both corn and soybean; CcrShv, cereal rye following corn, hairy vetch following soybean.

**Table 4 microorganisms-08-01773-t004:** Analysis of variance (ANOVA) results for the effects of cover cropping (CC), tillage (Till), and their interaction (CC × Till) on each group of principal components (PCs) extracted from the bacteria and fungi taxa data sets, which comprised bacterial and fungal indicator OTUs, respectively. The probability values of the treatment effects and degrees of freedom (df) are shown on the top rows. The treatment mean values and their standard errors are presented below. For each taxon and within a given column, treatment mean values followed by the same lowercase letter were not statistically different (α = 0.05).

		Bacteria							Fungi					
		PC1	PC2	PC3	PC4	PC5	PC6	PC7	PC1	PC2	PC3	PC4	PC5	PC6
Treatments	df	*p*-Value							*p*-Value					
Till	1	0.062	0.082	0.075	0.850	0.465	0.815	0.213	0.252	0.005	0.059	0.145	0.080	0.964
CC	2	0.004	0.009	0.045	0.231	0.485	0.128	0.173	0.873	0.517	0.044	0.001	0.419	0.796
CC × Till	2	0.968	0.101	0.131	0.363	0.014	0.025	0.285	0.682	0.339	0.128	0.497	0.039	0.171
Treatment means ^†^														
NT		−0.29	−0.22	−0.26	0.03	0.12	−0.03	−0.18	−0.16	−0.47b	−0.27	0.20	0.27	−0.01
T		0.29	0.22	0.26	−0.03	−0.12	0.03	0.18	0.16	0.47a	0.27	−0.20	−0.27	0.01
SEM		0.19	0.23	0.20	0.25	0.27	0.19	0.22	0.25	0.21	0.21	0.17	0.19	0.21
CT		−0.62b	0.43a	0.24a	0.24	−0.19	0.44	−0.19	0.09	−0.08	−0.23b	−0.72c	0.02	0.06
CarSar		−0.02a	0.10a	0.26a	−0.31	0.23	−0.16	0.38	0.00	−0.12	0.48a	0.69a	−0.23	−0.14
CcrShv		0.59a	−0.54b	−0.50b	0.07	−0.04	−0.27	−0.19	−0.09	0.21	−0.25b	0.03b	0.21	0.08
SEM		0.22	0.25	0.24	0.25	0.29	0.25	0.26	0.29	0.24	0.24	0.21	0.23	0.25
NT × CT		−0.87	0.44	−0.34	0.03	−0.39ab	0.24ab	−0.07	−0.23	−0.46	−0.70	−0.70	0.47ab	0.00
NT × CarSar		−0.27	−0.48	−0.02	−0.06	0.34ab	0.33a	0.13	−0.11	−0.84	0.04	1.06	−0.49b	0.23
NT × CcrShv		0.27	−0.64	−0.42	0.13	0.42a	−0.66b	−0.60	−0.14	−0.10	−0.15	0.23	0.83a	−0.25
T × CT		−0.36	0.42	0.82	0.46	0.01ab	0.64a	−0.30	0.41	0.29	0.25	−0.73	−0.43b	0.13
T × CarSar		0.32	0.68	0.54	−0.56	0.12ab	−0.66b	0.62	0.10	0.60	0.91	0.32	0.03ab	−0.51
T × CcrShv		0.92	−0.43	−0.58	0.00	−0.50b	0.11ab	0.22	−0.03	0.52	−0.35	−0.17	−0.41b	0.41
SEM		0.30	0.32	0.32	0.36	0.35	0.32	0.34	0.37	0.32	0.32	0.29	0.32	0.36

^†^ Tillage: NT, no-till; T, chisel tillage. Cover crops: CT, bare fallow control following both corn and soybeans; CarSar, annual ryegrass following both corn and soybean; CcrShv, cereal rye following corn, hairy vetch following soybean.

**Table 5 microorganisms-08-01773-t005:** Analysis of variance (ANOVA) results for the effects of cover cropping (CC), tillage (Till), and their interaction (CC × Till) on the relative abundances (RAs, %) of archaeal indicator OTUs. The probability values of treatment effects and the degrees of freedom (df) are shown on the top rows. The treatment mean values and their standard errors are presented below. Within a treatment, and for a given column, mean values followed by the same lowercase letter were not statistically different (α = 0.05).

		Archaea				
		C. Nitrososphaera	SCA1154	SCA1158	SCA1166	SCA1173
Factors	df	*p*-Value				
Till	1	0.533	0.236	0.181	0.107	0.248
CC	2	0.129	0.174	0.588	0.533	0.023
CC × Till	2	0.428	0.185	0.211	0.698	0.478
Treatment means ^†^						
NT		0.09	0.11	0.06	0.37	0.21
T		0.13	0.06	0.12	0.24	0.30
SEM		0.04	0.04	0.04	0.06	0.06
CT		0.03	0.04	0.12	0.30	0.40a
CarSar		0.17	0.05	0.07	0.37	0.13b
CcrShv		0.12	0.16	0.07	0.25	0.24ab
SEM		0.05	0.05	0.04	0.06	0.07
NT × CT		0.04	0.03	0.03	0.39	0.40
NT × CarSar		0.18	0.07	0.05	0.39	0.09
NT × CcrShv		0.06	0.23	0.08	0.32	0.15
T × CT		0.03	0.06	0.20	0.21	0.39
T × CarSar		0.17	0.04	0.09	0.35	0.18
T × CcrShv		0.19	0.09	0.06	0.18	0.33
SEM		0.06	0.06	0.05	0.09	0.09

^†^ Tillage: NT, no-till; T, chisel tillage. Cover crops: CT, bare fallow control following both corn and soybeans; CarSar, annual ryegrass following both corn and soybean; CcrShv, cereal rye following corn, hairy vetch following soybean.

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
