# Peer review of "Microbial Shifts Following Five Years of Cover Cropping and Tillage Practices in Fertile Agroecosystems"

_microorganisms, 2020, doi:10.3390/microorganisms8111773_

Round 1

Reviewer 1 Report

This reviewer found the study very interesting, well thought through and written in plain English. Work seems original, has significant content, scientific soundness and will be of interest to scientists in the field.

This may not be relevant changes - e.g. line 27 change "community composition" to "community compositions" throughout the text.

Author Response

Thank you for your positive review of our work. We have made the suggested change in line 27.

Reviewer 2 Report

This is a manuscript conducted to identify potential bioindicators of soil health from microbes responsive to cover crops and tillage treatments under field conditions. This study provides primary information on the soil microbiome within corn-soybean rotations including cover crops, which may help designing a targeted management strategy of the soil community composition with cover crops to reduce nutrient losses within the fertile conditions typical of prime agricultural landscapes. The experiment was well-designed, scientifically rigorous, and the manuscript was well-written. This would be of great interest to “microorganisms” readers. It is publishable with some revisions. Mainly, I think a better discussion of the soil chemical results would help improve the manuscript quite a bit.

For example, there was no significant effect of tillage and cover crop rotations on several soil properties (i.e., SOM, soil test-P, CEC, soil pH). To justify this lack of effect, the authors have only cited their own past studies [1, 2, 3]. However, they did not explain why reducing tillage intensity and improving cropping system diversity did not improve topsoil SOM and P, for example. There are several meta-analysis studies showing positive effects of those practices on soil health independent of soil classification (including Mollisols from the Midwest of the USA). Please, compare the results from your study to the results from those meta-analysis assessments. This discussion must be improved. 

  1. Behnke GD, Kim N, Villamil MB. Agronomic assessment of cover cropping and tillage practices across environments. Agron J. 2020:1-16.
  2. Dozier IA, Behnke GD, Davis AS, Nafziger ED, Villamil MB. Tillage and cover cropping effects on soil properties and crop production in Illinois. Agron J. 2017;109(4):1261-70.
  3. Necpalova M, Anex RP, Kravchenko AN, Abendroth LJ, Del Grosso SJ, Dick WA, et al. What does it take to detect a change in soil carbon stock? A regional comparison of minimum detectable difference and experiment duration in the north central united states. J Soil Water Conserv. 2014;69(6):517-31.

Another point is that there is no conclusion. The main goal was to identify potential bioindicators of soil health from microbes responsive to agricultural management practices under field conditions. What did you conclude? what do you recommend? 

Author Response

Thank you for your positive review of our manuscript, we truly appreciate your suggestions as we believe improve the quality and potential impact of our work.

In response to the first comment, we have added a paragraph l. 365 to 383 justifying the lack of effects on chemical properties with additional sources.

In response to the second comment, we have added a conclusion section l. 480 to 498 that details the bioindicators that we identified for cover cropping and tillage practices within these agroecosystems.

 Thank you again for your collegial service.